# Design of a Deployable Helix Antenna at L-Band for a 1-Unit CubeSat: From Theoretical Analysis to Flight Model Results

**DOI:** 10.3390/s22103633

**Published:** 2022-05-10

**Authors:** Lara Fernandez, Marco Sobrino, Joan Adria Ruiz-de-Azua, Anna Calveras, Adriano Camps

**Affiliations:** 1CommSensLab, Department of Signal Theory and Communications, UPC BarcelonaTech, 08034 Barcelona, Spain; sobrinohidalgo96@gmail.com (M.S.); adriano.jose.camps@upc.edu (A.C.); 2Department of Network Engineering, UPC BarcelonaTech, 08034 Barcelona, Spain; anna.calveras@upc.edu; 3Institute of Space Studies of Catalonia (IEEC), Space Science and Technology Research Group, CTE/UPC, 08034 Barcelona, Spain; 4i2Cat Foundation Space Communications Research Group, 08034 Barcelona, Spain; joan.ruizdeazua@i2cat.net

**Keywords:** CubeSat, nanosatellite, helix, deployable, antenna, L-Band, Earth Observation

## Abstract

The ${}^{3}$Cat-4 mission aims at demonstrating the capabilities of a CubeSat to perform Earth Observation (EO) by integrating a combined GNSS-R and Microwave Radiometer payload into a 1-Unit CubeSat. One of the greatest challenges is the design of an antenna that respects the 1-Unit CubeSat envelope while operating at the different frequency bands: Global Positioning System (GPS) L1 and Galileo E1 band (1575 MHz), GPS L2 band (1227 MHz), and the microwave radiometry band (1400–1427 MHz). Moreover, it requires between 8 and 12 dB of directivity depending on the band whilst providing at least 10 dB of front-to-back lobe ratio in L1 and L2 GPS bands. After a trade-off analysis on the type of antenna that could be used, a helix antenna was found to be the most suitable option to comply with the requirements, since it can be stowed during launch and deployed once in orbit. This article presents the antenna design from a radiation performance point of view starting with a theoretical analysis, then presenting the numerical simulations, the measurements in an Engineering Model (EM), and finally the final design and performance of the Flight Model (FM).

## 1. Introduction

Spaceborne Earth Observation (EO) data became available in the 1960s with the advances of artificial satellites. TIROS-1 [1] was the first EO satellite launched in 1960, and it embarked two black and white television cameras. Then, the TIROS program continued with four generations of satellites launched until 1994.

The EO satellite industry has evolved to make more complex spacecrafts, including multiple experiments on-board, such as Envisat [2] (2002), which is a European satellite that included eight different instruments, or the UARS [3] (1991) from the United States, which had 10 instruments. However, today’s trend is toward constellations of more dedicated satellite missions, such as the Sentinels [4], of the European Union Copernicus system.

In parallel, the so-called New Space industry has experienced rapid development of technologies for small satellites, especially since the appearance of the CubeSat standard [5] in 1999. This standard defines satellites by Units (U), where each U corresponds to 10 × 10 × 10 cm and can weigh up to 2 kg, as defined in the standard.

Among the EO payloads for CubeSats or small satellites, those that make use of Global Navigation Satellite System Reflectometry (GNSS-R) are gaining momentum, as they are passive (i.e., power consumption is small), the spatial resolution does not depend on the antenna size, and they allow performing ocean monitoring [6], obtaining soil moisture [7], and detecting flooded regions [8] or ice cover [9]. This technique has been validated in missions such as the UK-DMC [10], the UK TechnoDemoSat-1 (TDS-1) [11], and the CYGNSS constellation from NASA [12].

Additionally, combining GNSS-R with Microwave L-Band Radiometry (MWR) measurements improves the spatial resolution and accuracy of soil moisture and ocean salinity products [13]. This was done in the FSSCat mission [14] launched in 2020, in which the Flexible Microwave Payload 2 (FMPL-2) [15] combined both a GNSS-R and an RAD.

The ${}^{3}$Cat-4 mission [16] is a successor from the FSSCat mission. This mission aims at demonstrating similar capabilities (dual-band L1 and L2 GNSS-R and MWR) in a 1 Unit (1U) CubeSat. ${}^{3}$Cat-4 was selected by the ESA Fly your Satellite! 2 program, and it is planned to be launched in Q4 2022/Q1 2023 in the Ariane 6 maiden flight. The scientific payload is the Flexible Microwave Payload 1 (FMPL-1) [17], and it is based on an RTL-SDR. Additionally, an uplooking Right Hand Circular Polarization (RHCP) antenna and a downlooking Left Hand Circular Polarization (LHCP) L-Band Helix Antenna (LHA) also conform the FMPL-1. This downlooking antenna is included in the Nadir Antenna and Deployment Subsystem (NADS).

This work aims at defining the requirements necessary for the LHA. A preliminary theoretical analysis is first conducted to identify the dimensions of the antenna to meet the requirements. This analysis is conducted taking into consideration the constrains for the whole range of frequencies, instead of designing for one particular frequency, which is common. Then, it is verified with simulations, and a sensitivity analysis of the size and shape of the ground plane (GP) is performed to optimize the performance given the restrictive dimensions of a 1U CubeSat. Additionally, the simulations and the model are verified with measurements of an Engineering Model (EM) with different sized GPs; these measurements provide added value to the overall study, since current state of the art is just based on simulations. Finally, the design and measurements of the Flight Model (FM) are presented.

This article is organized as follows: Section 2 lists the LHA requirements. Section 3 contains the theoretical analysis of the design. Section 4 explains the results of the simulations performed, and it contains the GP sensitivity analysis. Section 5 includes the measurements of the EM model with different GPs. Section 6 presents the design and performance of the NADS FM. Finally, Section 7 summarizes the conclusions.

## 2. L-Band Helix Antenna Requirements Definition

To fulfill the mission goals, the LHA has to be directive, have a high front-to-back lobe ratio (FBR), LHCP, occupy less than 0.3 U, and cover a wide range of frequencies in the L-Band, from 1227 MHz (GPS L2) up to 1575 MHz (GPS L1). The overall requirements with values for each of the operating frequencies can be found in Table 1. These requirements have been defined by the Payload Engineer, and the scope of this work is to design an antenna compliant with them.

A common solution for GNSS-R payloads is to use an array of patch antennas that are placed on an external face of the satellite, as has been performned for TechDemoSat-1 [11], ${}^{3}$Cat-2 [18], CyGNSS [12], and FSSCat [14]. Other solutions consider different types of materials for transmission line-type antennas [19,20], although these have a similar performance to regular patch antennas. An array of any patch or transmission line-based antenna with the required directivity would be composed of six elements, needing a 6 U face to allocate the antennas. So, although this solution is simple in terms of mechanical interfaces, it is not valid for the ${}^{3}$Cat-4 mission, since it is a 1 U CubeSat.

Another solution is a helical antenna since it can be directive, broadband, and circularly polarized. These antennas are initially stowed in the satellite to then be deployed in orbit. Some missions have flown with commercial helical antennas, such as GOMX-1 [21] and GOMX-3 [22] as payload antennas, for ADS-B data acquisition. In addition, Lacuna Space [23] launched an Internet of Things (IoT) payload with a helical antenna on-board a Nanoavionics spacecraft. Moreover, there is a commercial solution for a quadrifilar helical antenna [24], but it occupies half a CubeSat Unit. Overall, helical antennas seem to be the best option for the ${}^{3}$Cat-4 mission.

There are also some studies on deployable antennas; in [25], a survey is performed on different types of antennas for CubeSats, and the equations and tools for the antenna design are provided. In addition, the design and testing on UHF [26] and VHF [27,28] antennas has been the object of research. In [29], there are also some large deployable antennas for CubeSats. Although there is literature on this topic, these solutions are not in the same frequency band and are not suitable for the ${}^{3}$Cat-4 mission due to the stringent space restriction (0.3 U). Moreover, none of the current solutions assess the impact that the size and shape of the ground plane has on the antenna directivity [30]. Thus, the LHA for ${}^{3}$Cat-4 requires a custom design.

## 3. Theoretical Background

The first step in the design process is to perform a theoretical analysis. This way, approximate values of the dimensions of the LHA are obtained, which can be verified through simulations at a later stage. The analysis presents novelty in the methodology it uses, since it considers the most constraining case from the range of frequencies, instead of designing just for a single frequency. To ease the understanding of the theoretical analysis, it is necessary first to define the reference names for the geometry of a helical antenna.

This geometry and reference names are shown in Figure 1, where *d* is the diameter of the helix, *N* is the number of turns, *S* is the spacing between turns, *C* is the circumference of the helix, *L* is the length of one turn, *∅* is the section of the coil, α is the pitch angle, *A* is the axial length of the deployed antenna, and *a* is the axial length of the stowed antenna. Out of these, the most relevant parameters are *d*, *N*, and *S*.

Helix antennas have three different radiation modes. This means that depending on the geometry of the antenna, the radiation pattern and directivity (D) change drastically. In Figure 2, the radiation modes can be seen. The normal mode (Figure 2a) radiates as a monopole, and it is achieved by significantly reducing S. In the case of the axial mode (Figure 2b), it radiates in the axial direction and provides high directivities. To achieve it, both the d and S have to be increased with respect to the normal mode. Finally, the conical mode (Figure 2c) has two lobes located at 45° with respect to the axial direction, and this mode is achieved by increasing significantly the diameter of the antenna.

Out of the three radiation modes, the most suitable one for the L-band Helix Antenna is the axial mode, since this mode ensures the maximum power density in the direction of its axis, and right or left hand circular polarization can be achieved depending on the direction of the antenna twisting. Thus, the LHA is designed following the axial mode geometry constraints.

The antenna radiates in the axial mode when the three conditions shown in Equation (Equation 1) are fulfilled. In the case of the LHA, these requirements have to be fulfilled for the whole range of frequencies.
(1)$$Axial\;mode=\left\{\begin{array}{c} 0.75<C/\lambda <1.33\\ 12{}^{\circ}<\alpha <15{}^{\circ}\\ N>3\;\mathrm{turns} \end{array}\right.$$

Taking into account the geometry of an unrolled antenna turn seen in Figure 1b, it can be obtained through trigonometry that tan(α)=S/C. In addition, Equation (Equation 1) must be fulfilled for the whole frequency range (Table 2) by using the highest wavelength value for lower bounds and the lowest wavelength value for upper bounds, as seen in Equation (Equation 2).
(2)$$Axial\;mode=\left\{\begin{array}{c} 0.75\lambda_{1}<C<1.33\lambda_{2}\\ tan(12{}^{\circ})C<S<tan(15{}^{\circ})C\\ N>3\;\mathrm{turns} \end{array}\right.$$

Finally, substituting the wavelength values in Equation (Equation 2), the upper and lower bounds for each condition are obtained:(3)$$Axial\;mode=\left\{\begin{array}{c} 183.3\;\mathrm{mm}<C<247.5\;\mathrm{mm}\\ 39.28\;\mathrm{mm}<S<66.08\;\mathrm{mm}\\ N>3\;\mathrm{turns} \end{array}\right.$$

The dimensions of the antenna (Table 3) have been chosen to be inside that range. First, by fixing the spacing between turns (S) to the middle of the range, the *d*, α, *C*, and *L* values are obtained. Moreover, the twisting of the helix is assumed to be toward the left, obtaining LHCP.

Then, the parameters that are not determined by *S* are the *N*, *A*, *a*, and *∅*. For *N*, it is fixed based on the *D*; thus, this value is augmented until the *D* requirements are achieved. A margin of safety of 20% has been considered since the theoretical approximation of *D* is optimistic with respect to reality. This value has been taken from previous experience, comparing theoretical directivities with simulation results.

The D is computed as shown in Equation (Equation 4):(4)$$D=\frac{4\pi }{\int_{0}^{2\pi }\int_{0}^{\pi /2}t_{N}\left(\theta ,\phi \right)\cdot sin\theta \delta \theta \delta \phi }$$
where the normalized radiation pattern is calculated as:(5)$$t_{N}\left(\theta ,\phi \right)=\frac{|E_{N}{\left(\theta \right)|}^{2}}{max(|E_{N}\left(\theta \right){|}^{2})}$$
and the electric field as:(6)$$|E_{N}{\left(\theta \right)|}^{2}=|sin\left(\frac{\pi }{2N}\right)\cdot cos\theta \cdot \frac{sin(N\psi /2)}{sin(\psi /2)}{|}^{2};\;where\;\psi =k\cdot S\left(cos\theta -1\right)-\frac{\pi }{N}$$

The results for the *D* can be seen in Table 4. As it can be seen, the *D* with the margin of safety fulfills the requirements for the L-Band Helix Antenna, and these requirements correspond to N=11 turns.

Having *N* fixed, the axial length can be computed as A=N·S. Thus, the final parameter left to define is the section of the coil. It has been chosen to be 1 mm, since the antenna must be flexible, to be easily stowed.

## 4. Simulation Results

This section contains the simulation results performed with the antenna design defined in Section 3. Along with the different simulations, the impact of the GP is assessed, and a solution that complies with the LHA requirements (D, FBR, and GP dimensions and shape) is presented.

### 4.1. Assumptions and Modeling

The antenna and the CubeSat structure have been modeled using SolidWorks and the electromagnetic behavior with CST Studio Suite using the Frequency Solver. To perform the simulations, certain assumptions have been made when modeling the antenna. These are listed below:The antenna has the measurements specified in Section 3.The ground planes have a thickness of 35 μm.The ground plane’s material is copper.The antenna’s material considered is Steel 1008.The antenna is fed emulating a connector conformed on the outside by the ground plane, an FR-4 dielectric insulator, and the antenna wire as the inner connector.The antenna is fed using a waveguide port.Other subsystems in the satellite are not considered in the model.

### 4.2. Ground Plane Impact Analysis

Initially, two different cases are studied. One considers an infinite GP, as seen in Figure 3. According to [31], a GP can be considered “infinite” if it is larger than 2λ, although for the study, a GP of 10λ by 10λ (217.4 mm) has been simulated. This case is considered as the reference value for the other model, which is just 94 mm by 90 mm GP, (Figure 4), which is the space available for the GP inside a 1 U CubeSat.

The simulation results are shown in Table 5. First comparing the results from the infinite GP to the theoretical directivities obtained in Table 4, it can be seen that the simulated values are lower than the theoretical ones. This was expected since theoretical calculations do not consider the different materials, the losses, or the size of the GP.

Moreover, comparing the simulated results between the infinite and CubeSat-sized GPs, it can be seen that the lower the frequency (i.e., the higher the wavelength), the larger the effect it has on the antenna performance both in terms of D and FBR. Taking a closer look at the D, it can be seen how at 1227.6 MHz, it is 2.01 dB less than the CubeSat-sized GP. For the MWR band, the D is the same, while for GPS L1, it has 0.2 dB more. In the case of the FBR, it is mostly affected by the reduction in the size of the GP. Thus, alternative GP solutions have been explored to assess if the FBR improves.

### 4.3. Ground Plane Shapes Analysis

Following [32], four alternative GPs are simulated: (1) an infinite GP, as used in the previous analysis; (2) a square GP (Figure 5a); (3) a cylindrical GP (Figure 5b); and (4) a conical GP (Figure 5c). For each of these configurations, the optimum dimensions identified in the reference are used and are compared with the infinite GP.

“Infinite” ground plane: For this configuration, the same assumption as in the previous section has been taken of 10λ by 10λ. For the rest of this section, it will be referred to as 10λ.Square ground plane: This plane is calculated considering two different measurements of the side (b), one with b=0.5λ and the other with b=1.5λ, which should give more D and less FBR.Cylindrical cup ground plane: This plane should provide more D than the square GP. According to [32], the optimum dimensions are $d_{gp}=\lambda$ and h=0.25λ (Figure 6d).Conical ground plane: This GP should act as a reflector and provide the largest D of all other GPs. The optimum dimensions are $d_{gp1}=0.75\;\lambda$, $d_{gp2}=2.5\;\lambda$ and h=0.5λ (Figure 6c).

**Figure 6 sensors-22-03633-f006:**
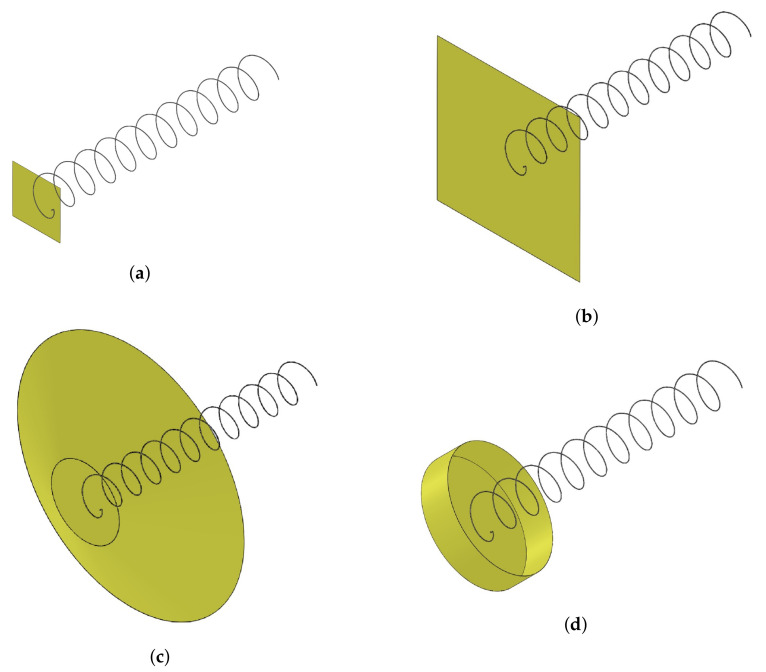
Analysis ground plane models. (**a**) Square model with 0.5 λ GP. (**b**) Square model with 1.5 λ GP. (**c**) Model with conical GP. (**d**) Model with cylindrical GP.

The results for these analyses are shown in Figure 7a,b, where the first one represents a comparison of the D, and the second one represents the FBR.

Comparing the results for the D, several conclusions can be drawn. Regarding the 10λ GP, it can be seen that it does not maximize the D, since other smaller GPs can provide even higher Ds. Regarding the 0.5λ square GP, the D is lower for low frequencies and equal to the infinite GP for higher. For the 1.5 λ GP, the D is higher than for the 10λ GP. The cylinder has higher D for frequencies above 1300 MHz is even higher than for the 1.5 λ. Finally, the conical GP has even more D than the theoretical one.

Comparing the FBR, it can be seen that the 0.5λ GP does not provide enough FBR to meet the requirements. The other GPs do comply with the requirements; in the case of the 1.5λ, the results provide 15 dB of FBR for the whole range, and for the cylindrical, the results are even better having around 20 dB of FBR. The 10λ GP has between 35 and 25 dB of FBR, and for the conical GP, the FBR is the highest in almost the whole frequency band.

Overall, the GPs that comply with both D and FBR requirements are the 1.5λ square GP, the cylindrical, the conical, and the 10λ. However, all of these GP are not able to fit in a 1U CubeSat.

Out of all of these, the cylindrical GP is easier to integrate in a smaller dimension within the CubeSat structure. However, given that the structure of the CubeSat is squared-shaped, it would be ideal and even easier to integrate a cubic cup GP. For this reason, in the following subsection, a comparison between the cylindrical and the cubic cup GPs is made.

### 4.4. Cylindrical and Cubic Cup Ground Plane Comparison Analysis

As identified previously, the ideal dimensions for the Cylindrical cup GP are $d_{gp}=\lambda$ and h=0.25λ. However, in the CubeSat, these dimensions do not fit, the maximum dimensions that can fit are $d_{gp}$ = 96 mm ≈ 0.45 λ and *h* = 20 mm ≈ 0.1 λ. For this reason, the cylindrical cup GP is simulated for different $d_{gp}$ and h sizes, and the D is studied.

For the analysis, first, the $d_{gp}$ is altered between $0.4\;\lambda \leq d_{gp}\leq \lambda$, leaving the optimum h=0.25λ. Then, the h is altered between 0.1λ≤h≤0.5λ, leaving the optimum $d_{gp}=\lambda$.

The results for these analyses are shown in Figure 8a,b, being the first sweep in $d_{gp}$ and the second in *h*. For the case of the $d_{gp}$, it can be seen that as D reduces, the GP size is also reduced. For the specific case of $d_{gp}=0.45\;\lambda$, the D is decreased by approximately 3 dB in all bands with respect to the maximum D. For h=0.2λ, the highest D is obtained, and for h=0.1λ, D decreases approximately 0.5 dB.

Overall, these simulations show that the CubeSat dimensions are actually restricting the antenna performance. However, the performance is better than for other GPs that can be fitted within the CubeSat.

The results presented in Table 6 show that the Cubic Cup GP provides higher D in all bands and the FBR is lower in GPS L2 but higher for the rest of the bands. Although this Cubic Cup GP does not comply with the FBR requirement for the GPS L2 band (Table 8), this is the GP configuration selected for the final CubeSat design because of the better performance for the rest of the parameters.

### 4.5. CubeSat Structure Impact Analysis

This final analysis plans to assess the impact that the CubeSat structure itself has on the antenna performance. This is done by simulating the Cubic Cup GP standalone (Figure 9), and with the CubeSat structure and solar panels, (Figure 10).

The results for these simulations are shown in Table 7. As it can be seen, the results both in D and FBR appear to be worse for the model with the CubeSat structure. However, it would be better to assess this in a prototype with real measurements, since the structure has a complex geometry, and this is not often well parametrized by the simulation software. Finally, a summary of the requirements compliance with all models simulated along this section can be found in Table 8.

## 5. Engineering Model Results

This section contains the measurements in an Anechoic Camber from the key GP shapes identified in Section 4. These measurements were done to verify the simulation model and to assess the best suited configuration for the LHA. In total, four different measurements were performed: (1) “infinite” GP, (2) CubeSat-sized GP, (3) Cubic Cup GP, and (4) Cubic Cup GP with CubeSat structure.

### 5.1. Engineering Model

The antenna has been prototyped using a 3D printed mold, with a wire running along. This way, the measurements identified in Table 3 can be ensured. Along with the measurement campaign, different GP were soldered to this antenna. These GPs were manufactured with either aluminum plates or with 1.6 mm thick FR-4 printed circuit boards (PCBs) with a 35 μm copper metallization.

These prototypes are measured in the UPC AntennaLab Anechoic Chamber [33]. The measurements are performed at 1227 MHz, 1400 MHz, 1427 MHz, and 1575.42 MHz.

### 5.2. Infinite Ground Plane Measurements

The first measurement performed was with an “infinite” GP, although this was modeled in the simulation as a square with 10λ side. In reality, the GP has a side of 2λ, since 10λ was too large to be manufactured, handled, and adapted to the Anechoic Chamber rotor. This GP has been made out of an aluminum plate.

The results can be seen in Table 9. Comparing the results with the simulation from Table 5, in terms of D, the results are similar with variations of a maximum of 1 dB. For the FBR, the measured results provide higher FBR than the simulations, complying with the requirements. However, this GP does not fit inside the CubeSat structure, but these results are taken as a reference to compare with the other measurements.

### 5.3. CubeSat-Sized Ground Plane Measurements

For this measurement, the antenna has been assembled with a CubeSat-sized GP. The GP has the same dimensions considered in the simulations of 94 by 90 mm, and it is made with an FR-4 PCB.

Comparing these results (Table 10) with the simulations in Table 5, the D is lower than in the simulations by a maximum of 1.90 dB. In the case of the FBR, the results for the measurements are better than the simulation. However, neither D nor FBR are high enough to meet the requirements of the LHA. In addition, as expected, the results are worse than the ones obtained with the infinite GP.

### 5.4. Cubic Cup Ground Plane Measurements

For this measurement, a cubic cup GP as in the simulations has been assembled to the antenna. This GP is made out of FR4 PCBs, with the measurements identified as 94 mm by 90 mm and a height of 20 mm.

The results can be seen in Table 11. Comparing them with the simulations, in terms of D, the values are 1 dB lower for the real model. However, the FBR has values up to 6 dB higher. If the results are compared to the measurements of the CubeSat-sized GP, D is at least 1 dB higher for all frequencies, and the FBR is significantly higher for the GPS L2 band. Overall, the impact of the Cubic Cup GP benefits the antenna’s performance, and it complies with all requirements for the LHA.

### 5.5. Cubic Cup Ground Plane with Cubesat Structure Measurements

Given that the cubic cup GP complies with all requirements and it provides good results, one additional measurement has been made. The model for this final measurement is the antenna with the cubic cup GP inside the CubeSat structure, as seen in Figure 11.

Results are shown in Table 12. Comparing these results with the ones without the CubeSat structure (Table 11), it can be seen that all values are similar, meaning that the structure does not have a major effect on the antenna performance.

## 6. Flight Model Results

In this section, the design and performance of the LHA Flight Model (FM) are presented, detailing the final design, matching, and radiation pattern performance.

### 6.1. Design

The Flight Model of the LHA is included within the Nadir Antenna and Deployment Subsystem (NADS). The NADS not only includes the antenna but also contains the deployment mechanism. A detailed description of this deployment mechanism and the qualification tests done to verify its correct functionality is available in [34]. Moreover, a characterization of the deployment and the effect it has on the satellite can be found in [35].

The design of the NADS is presented in Figure 12. The different elements of the subsystem and their functionalities are explained below:Aluminum shield: this part serves both as GP and as a mechanical support for the deployment mechanism.Deployment board: this PCB contains the circuitry for the antenna deployment.Brass fingers: these hold the antenna in stowed configuration.Helix antenna: is the designed radiating element.Fabric sheath: this fabric holds the shape of the antenna and provides rigidity.Stage board: PCB used to hold the antenna halfway through the deployment.Gravity boom: this part first helps to hold the antenna compressed by providing a surface where the fingers lock, and it also acts as a boom for the satellite’s attitude control. It also slightly increases the directivity because of its dielectric constant.Boom holder: this piece holds the fingers locked to the gravity boom and prevents overall movement to the antenna when stowed. 

The dimensions of the subsystem are 87 mm × 87 mm × 27.5 mm stowed, and 87 mm × 87 mm × 515 mm deployed. In the end, the GP had to be slightly reduced, since it was not possible to integrate the 94 by 98 mm GP.

The materials, manufacturing, and surface treatments used for the different parts of the NADS are listed in Table 13. The most relevant ones for this particular case are the aluminum shield, which is made out of aluminum 7075 and has a surtec 650 treatment to ensure electrical conductivity. In addition, the antenna has been made out of spring steel, and a tin surface treatment has been applied so it can be soldered to the output connector.

### 6.2. Antenna Matching

The matching of helical antennas is traditionally done by inserting copper or other metallic pieces between the first turn of the antenna and the GP. This methodology changes the input resonance impedance of the antenna, and it can be matched to 50 by adding or removing the quantity of the material.

Although this methodology is easy, fast, and it can be done by trial and error, it is not suitable for the NADS, because it can be damaged when the antenna is stowed. Instead, the following methodology has been implemented.

The NADS matching methodology is inspired by [36], where it is suggested to leave part of the last turn of the antenna parallel to the GP. In the NADS, it would be difficult to ensure this spacing between the antenna and the GP, especially after stowing and deploying the antenna. Therefore, an alternative solution had to be taken.

The final solution has been to place a 1 mm thick FR-4 PCB between the antenna and the GP to ensure this distance is always constant. The matching is tuned by sewing part of this last turn to the FR-4 PCB. This matching process was first performed in the Qualification Model (QM), where up to the first half of the antenna was sewed to the FR-4 PCB. Finally, the best results were obtained sewing a quarter of a turn.

These results measured in a Vector Network Analyzer (VNA) are presented in Figure 13, showing around −20 dB of S11 in the GPS L2 band, −12 dB in the Radiometer band, and −8.5 dB in the GPS L1 band. This last value for GPS L1 has been considered as valid, even being higher than the −10 dB common value, which is the best matching value that could be obtained. Given that the range of frequencies in which the antenna has to operate is quite broad, the process was extremely challenging.

### 6.3. Radiation Pattern

The radiation pattern of the NADS FM was also measured in the UPC Antenna Lab Anechoic Chamber [33]. A picture of this subsystem integrated in the chamber’s rotor is shown in Figure 14.

Measured results are shown in Table 14 and the radiation patterns in Figure 15. These results are compliant with the requirements presented, and they are similar to the ones with the engineering model (Table 12). Finally, in this measurement, the gain was also measured, and it shows that the antenna is correctly matched, as expected from the results shown in Figure 13.

The axial ratio (AR) for LHCP was also measured in the NADS FM. The results (Figure 16) show that the antenna has AR approximately 0 dB from −30° up to 30° in all frequencies. This complies with the LHCP requirements, since an antenna is considered circularily polarized as long as the axial ratio is lower than 3 dB.

## 7. Conclusions

This study has presented the design, electromagnetic simulation, prototyping, and FM results of the LHA downlooking antenna of the FMPL-1 GNSS-R and MWR payload. This antenna will fly on-board the NADS Subsystem of the ${}^{3}$Cat-4 mission which will be launched in the Ariane 6 maiden flight.

The first step of the study was the theoretical antenna design; from this, the dimensions of the antenna have been obtained. With these dimensions, a sensitivity analysis for different GP shapes and dimensions has been conducted. Among the different GPs analyzed, the “infinite” GP 10λ has been taken as the reference, at the end, the GP that provides more performance and can fit inside a CubeSat structure is a cubic cup GP. This GP provides 9.88 dB of D and 6.78 dB of FBR for the GPS L2 band, above 11.80 dB of D and 11.71 dB of FBR in the mwRAD band, and 13.60 dB of D and 15.70 dB of FBR for GPS L1.

Following the simulations, a prototype of the antenna has been manufactured and measured with different shapes of GP. The final shape, also given by the good performance in the simulations, has been the same cubic cup GP identified in the simulations. The results of this GP measured with the CubeSat structure were 8.78 dB of D and 12.72 dB of FBR for the GPS L2 band, above 11.75 dB of D and 17.25 dB of FBR in the mwRAD band, and 13.49 dB of D and 24.08 dB of FBR for GPS L1.

Finally, the FM has also been presented, with the final design, matching, and radiation pattern, satisfying all the requirements. The values obtained from the FM were 9.86 dB of D and 22.65 dB of FBR for the GPS L2 band, above 11.26 dB of D and 28.42 dB of FBR in the mwRAD band, and 12.11 dB of D and 19.87 dB of FBR for GPS L1.

## Figures and Tables

**Figure 1 sensors-22-03633-f001:**
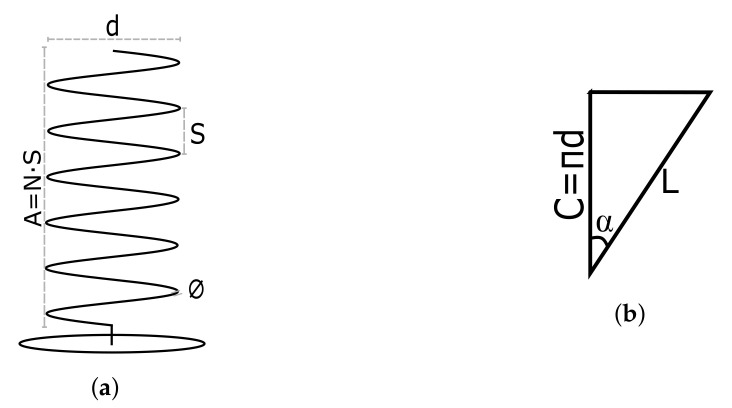
Overall geometry of helical antennas. (**a**) Geometry of helical antenna. (**b**) Unrolled turn of helical antenna.

**Figure 2 sensors-22-03633-f002:**
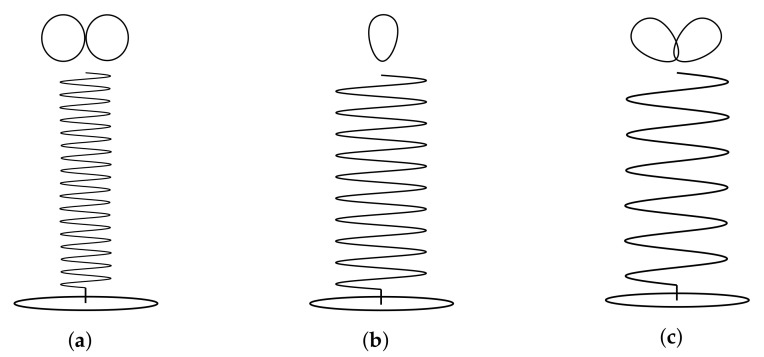
Helix antenna radiation modes. (**a**) Normal mode. (**b**) Axial mode. (**c**) Conical mode.

**Figure 3 sensors-22-03633-f003:**
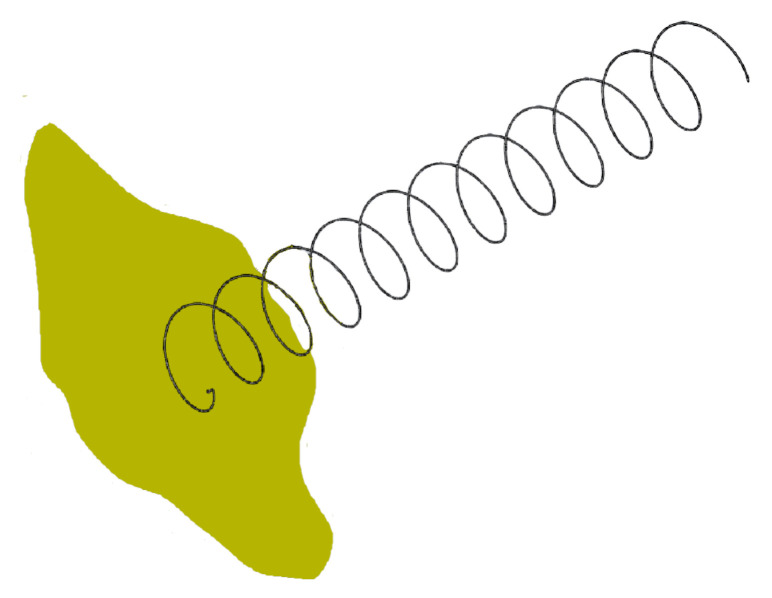
Model with infinite GP.

**Figure 4 sensors-22-03633-f004:**
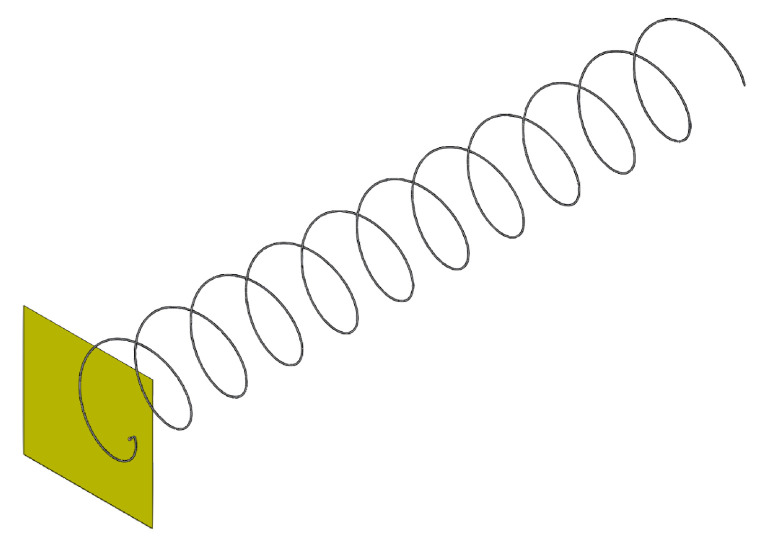
Model with CubeSat GP.

**Figure 5 sensors-22-03633-f005:**
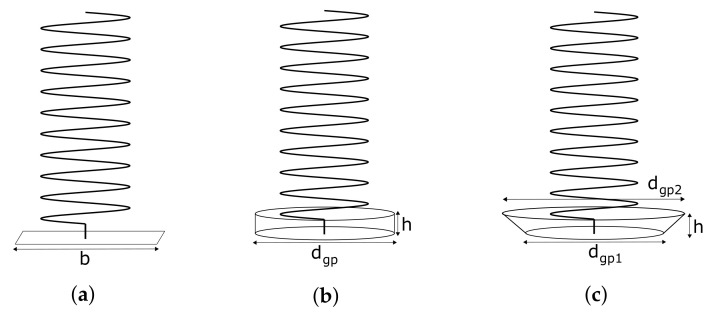
Ground plane models and dimension parameters. (**a**) Square GP. (**b**) Cylindrical GP. (**c**) Conical GP.

**Figure 7 sensors-22-03633-f007:**
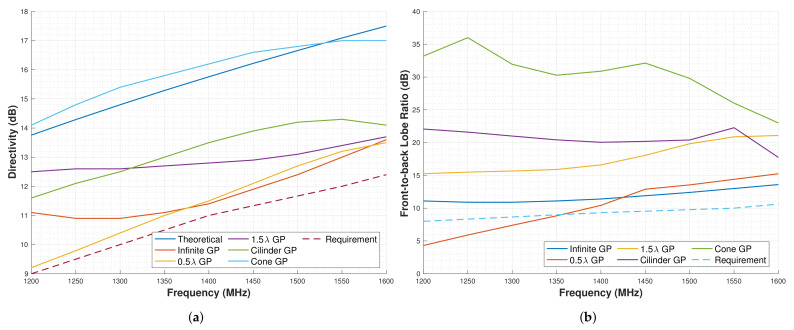
Results for the ground planes shapes analysis. (**a**) Directivity results for ground plane shapes analyzed. (**b**) Front-to-back lobe ratio results for ground plane shapes analyzed.

**Figure 8 sensors-22-03633-f008:**
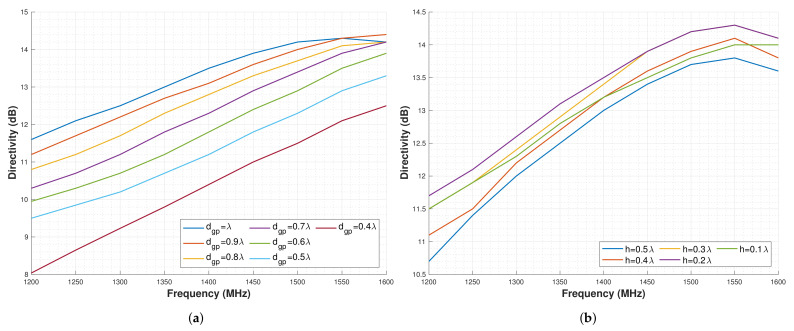
Directivity results for a sweep in parameters for the cylindrical GP. (**a**) Directivity for different cylindrical GP diameters and optimum height (h=0.25λ). (**b**) Directivity for different cylindrical GP heights and optimum diameter ($d_{gp}=\lambda$).

**Figure 9 sensors-22-03633-f009:**
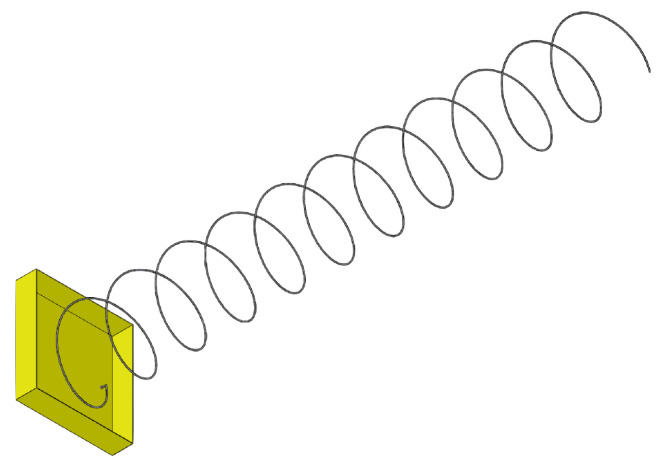
Model with cubic cup ground plane.

**Figure 10 sensors-22-03633-f010:**
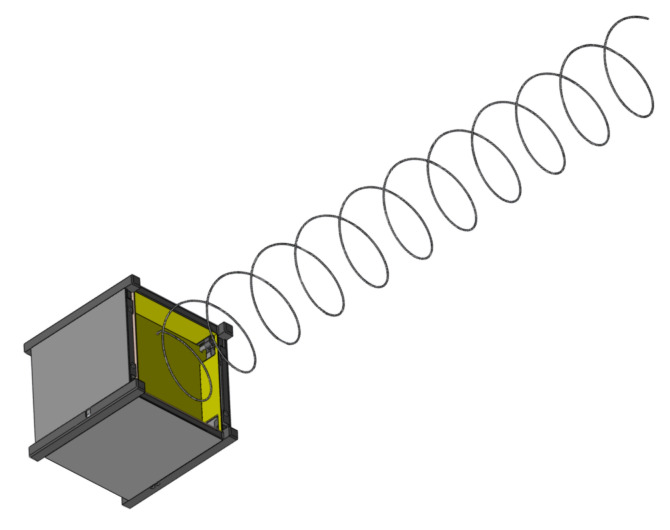
Model with cubic cup ground plane and CubeSat structure.

**Figure 11 sensors-22-03633-f011:**
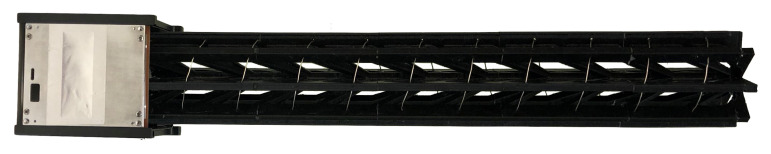
Prototype assembled in CubeSat structure.

**Figure 12 sensors-22-03633-f012:**
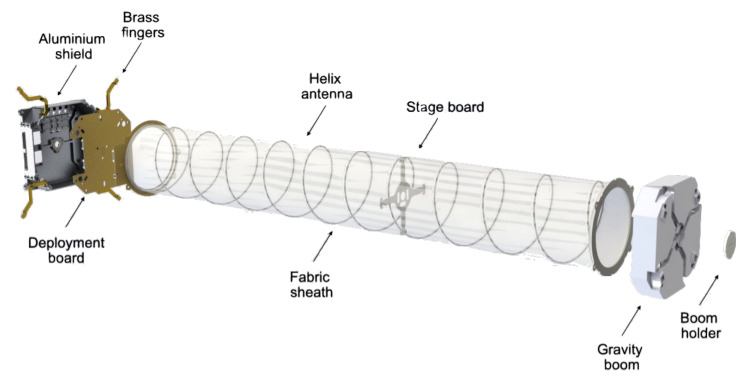
Extruded view of the NADS.

**Figure 13 sensors-22-03633-f013:**
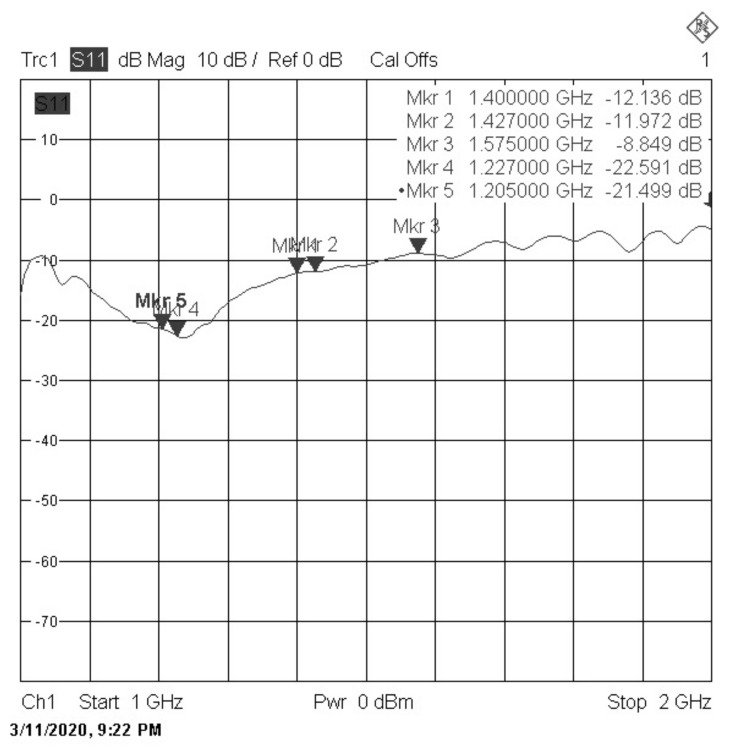
S11 parameters results for the NADS FM.

**Figure 14 sensors-22-03633-f014:**
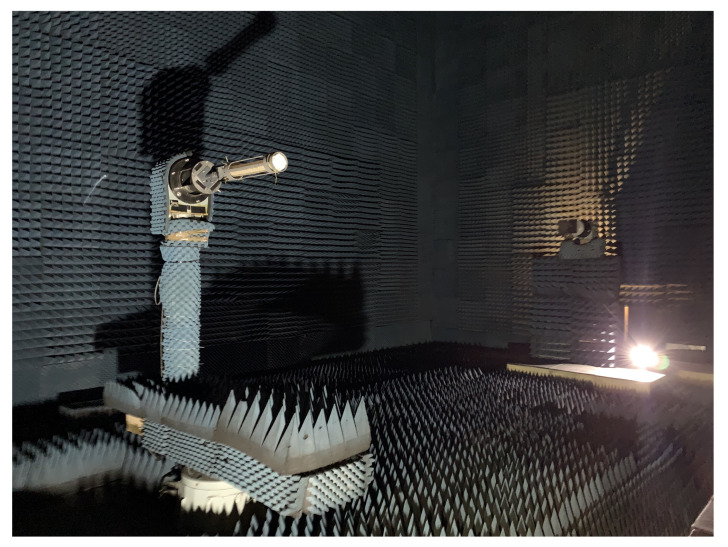
NADS FM in the UPC Antenna Lab Anechoic Chamber.

**Figure 15 sensors-22-03633-f015:**
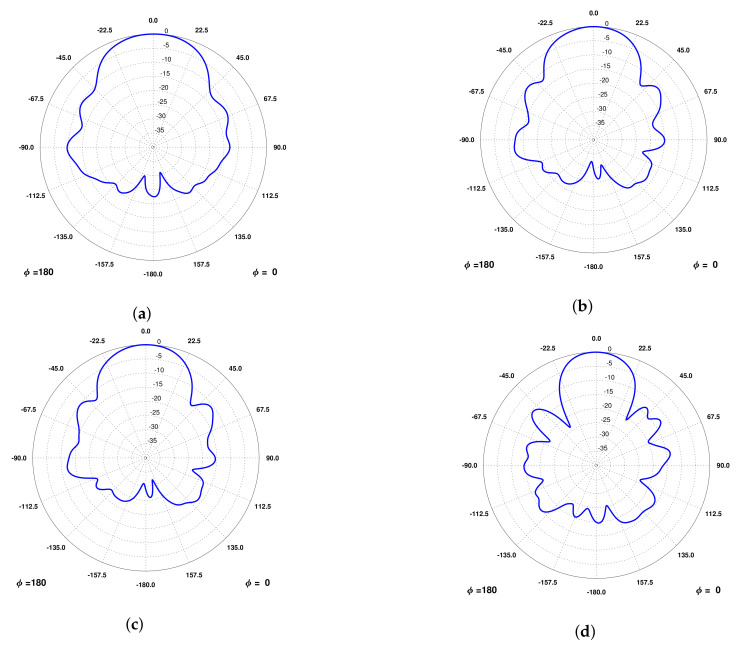
Radiation patterns measured in an Anechoic Chamber from the NADS Flight Model. (**a**) Radiation at 1227.6 MHz. (**b**) Radiation at 1400 MHz. (**c**) Radiation at 1427 MHz. (**d**) Radiation at 1575 MHz.

**Figure 16 sensors-22-03633-f016:**
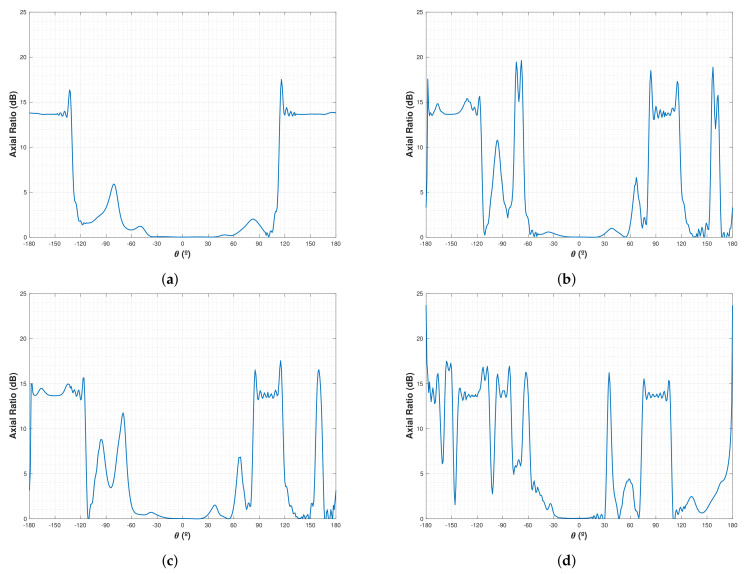
Axial ratio measured in an Anechoic Chamber from the NADS Flight Model. (**a**) Axial ratio at 1227.6 MHz. (**b**) Axial ratio at 1400 MHz. (**c**) Axial ratio at 1427 MHz. (**d**) Axial ratio at 1575 MHz.

**Table 1 sensors-22-03633-t001:** Requirements for the LHA.

ID	Requirement Description
001	The L-Band antenna shall work at 1575.42 MHz (GPS L1 and Galileo E1 band) with a minimum directivity of 12 dBi
002	The L-Band antenna shall have a minimum front-to-back lobe ratio of 10 dB at 1575.42 MHz
003	The L-Band antenna shall work in the radiometry band between 1400 and 1427 MHz with a minimum directivity of 11 dBi
004	The L-Band antenna should work at 1227.60 MHz (GPS L2 band) with a minimum directivity of 9 dBi
005	The L-Band antenna should have a minimum front-to-back lobe ratio of 8 dB at 1227.60 MHz
006	The L-Band antenna shall receive in Left Hand Circular Polarization (LHCP) for the L1, E1 and L2 bands
007	The L-Band antenna shall be matched to 50 impedance with a minimum of 12 dB of reflection losses for Radiometry band (1400–1427 MHz)
008	The L-Band antenna shall have an stowed envelope of at maximum 0.3 U

**Table 2 sensors-22-03633-t002:** Design frequency range.

Band	Frequency (MHz)	Wavelength (mm)
GPS L2	$f_{1}$ = 1227.6	$\lambda_{1}$ = 244.4
GPS L1	$f_{2}$ = 1575.42	$\lambda_{2}$ = 190.4

**Table 3 sensors-22-03633-t003:** Design parameters of the antenna.

**d**	61.8 mm	∅	1 mm	**N**	11 turns
α	12.36°	**S**	46 mm	**C**	213.94 mm
**L**	214.87 mm	**A**	506 mm	**a**	11 mm

**Table 4 sensors-22-03633-t004:** Calculated theoretical directivities.

Frequency (MHz)	Theoretical D (dB)	D with 20% Margin
1227.6	13.76	11.00
1400	15.48	12.38
1427	15.73	12.58
1575.42	17.02	13.61

**Table 5 sensors-22-03633-t005:** Simulation results for Infinite and CubeSat-sized ground planes.

Band	Freq	Infinite GP		CubeSat GP	
	(MHz)	D (dB)	FBR (dB)	D (dB)	FBR (dB)
GPS L2	1227.6	10.90	33.90	8.89	3.24
RAD	1400	11.80	32.35	11.30	9.00
RAD	1427	12.10	30.95	12.10	11.10
GPS L1	1575.42	13.10	33.63	13.30	13.76

**Table 6 sensors-22-03633-t006:** Simulation results for Cylindrical Cup and Cubic Cup ground planes.

Band	Freq	Cylindrical Cup GP		Cubic Cup GP	
	(MHz)	D (dB)	FBR (dB)	D (dB)	FBR (dB)
GPS L2	1227.6	9.15	11.05	9.88	6.78
RAD	1400	11.40	10.02	11.80	11.71
RAD	1427	11.60	10.04	12.10	12.66
GPS L1	1575.42	13.30	11.13	13.60	15.70

**Table 7 sensors-22-03633-t007:** Simulation results for cubic cup CubeSat sized ground plane.

Band	Freq	Cubic Cup GP		Cubic Cup GP CubeSat	
	(MHz)	D (dB)	FBR (dB)	D (dB)	FBR (dB)
GPS L2	1227.6	9.88	6.78	7.41	4.18
RAD	1400	11.80	11.71	11.30	11.57
RAD	1427	12.10	12.66	11.70	12.31
GPS L1	1575.42	13.60	15.70	13.60	17.03

**Table 8 sensors-22-03633-t008:** Requirements compliance for the GP shapes simulated along Section 4.

	Requirement ID							
	**1**	**2**	**3**	**4**	**5**	**6**	**7**	**8**
Infinite GP	✓	✓	✓	✓	✓	✓	-	✗
CubeSat GP	✓	✓	✓	✗	✗	✓	-	✓
0.5λ square GP	✓	✓	✓	✓	✗	✓	-	✗
1.5λ square GP	✓	✓	✓	✓	✓	✓	-	✗
Cylindrical GP	✓	✓	✓	✓	✓	✓	-	✗
Conical GP	✓	✓	✓	✓	✓	✓	-	✗
CubeSat-sized cylindrical GP	✓	✓	✓	✓	✓	✓	-	✓
Cubic Cup GP	✓	✓	✓	✓	✗	✓	-	✓

**Table 9 sensors-22-03633-t009:** Measurements of the antenna with infinite ground plane.

Band	Freq (MHz)	Directivity (dB)	FBR (dB)
GPS L2	1227.6	11.28	43.45
RAD	1400	10.80	38.43
RAD	1427	11.27	37.85
GPS L1	1575.42	12.47	47.77

**Table 10 sensors-22-03633-t010:** CubeSat-sized ground plane measurements.

Band	Freq (MHz)	Directivity (dB)	FBR (dB)
GPS L2	1227.6	7.49	5.70
RAD	1400	9.43	15.46
RAD	1427	10.52	17.39
GPS L1	1575.42	11.44	16.80

**Table 11 sensors-22-03633-t011:** Cubic cup ground plane measurements.

Band	Freq (MHz)	Directivity (dB)	FBR (dB)
GPS L2	1227.6	8.89	11.56
RAD	1400	10.74	17.92
RAD	1427	11.78	18.93
GPS L1	1575.42	12.55	25.67

**Table 12 sensors-22-03633-t012:** Cubic cup ground plane with structure measurements.

Band	Freq (MHz)	Directivity (dB)	FBR (dB)
GPS L1	1227.6	8.78	12.72
RAD	1400	12.00	18.46
RAD	1427	11.85	17.25
GPS L2	1575.42	13.49	24.08

**Table 13 sensors-22-03633-t013:** Materials of NADS parts.

Part	Manufacturing	Material	Surface Treatment
Aluminum shield	5 axes CNC	Aluminum 7075	Surtec 650 Alodine
Brass fingers	Casting	Brass	-
Deployment board	Chemical process	FR-4	-
Helix antenna	Helical extrusion	Spring steel	Tin
Fabric sheath	Laser cut	PTFE fabric	-
Stage board	Chemical process	FR-4	-
Gravity boom	5 axes CNC	PTFE	-
Boom holder	5 axes CNC	PTFE	-

**Table 14 sensors-22-03633-t014:** Results from the measurements in an Anechoic Chamber of the NADS Flight Model.

Band	Freq (MHz)	Directivity (dB)	Gain (dB)	FBR (dB)
GPS L2	1227.6	9.86	8.89	22.65
RAD	1400	11.26	10.42	28.42
RAD	1427	11.44	10.60	29.02
GPS L1	1575.42	12.11	11.15	19.87

## Data Availability

Not applicable.

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
