# Peer review of "Design of a Deployable Helix Antenna at L-Band for a 1-Unit CubeSat: From Theoretical Analysis to Flight Model Results"

_sensors, 2022, doi:10.3390/s22103633_

Round 1

Reviewer 1 Report

This manuscript looks like an engineering report rather than an innovative paper, which includes some aspects as follows:

  1. The whole story tells the process from indicators to antenna testing, which is within the range of conventional helical antenna design and do not show the innovation of this manuscript.
  2. For abstracts, more than half of the paragraph is talk about the background of the study, the rest talks about the indicators, there is less description of the specific work for this manuscript.
  3. The theoretical background in Chapter 3 is unnecessary to describe in detail, unless this part is innovative, and please highlight it.
  4. From Fig.14, Mark 3 for 1.575 GHz does not have good impedance match, which is not below the -10 dB as generally identified in the industry. Besides that, Fig. 14 needs to be redrawn for aesthetics and readability.
  5. Test environment photos don't need to be shown twice, for Fig. 11 and Fig .15.
  6. As a circularly polarized antenna, the discussion of the axis ratio is essential, please add the relevant simulation and measurement results.
  7. Fig. 16 has many drawing problems, please check.

Reviewer 2 Report

Authors in this research work have presented and investigated the design, electromagnetic simulation, prototyping, and FM results of the LHA downlooking antenna of the FMPL-1 GNSS-R and MWR payload. This antenna will fly on-board the NADS Subsystem of the 3Cat-4 mission which will be launched in the Ariane 6 maiden flight. The topic and concept of this work were found interesting. The promising results have been achieved and properly discussed in a well-organized manuscript. The results have been validated by providing the fabricated prototype. Although this work seems to be attractive for antennas and sensors societies, authors are requested to carefully revise it by properly addressing the following comments to improve its quality prior to final recommendation.

1) Abstract section which includes some references as well is more similar to an introduction part. Authors only in the last sentence have provided very brief information about the proposed antenna. So, this section should be totally removed and replaced with a proper abstract including below information. So, the current abstract section can be added in the introduction section.

a) Please first provide the design process of the proposed deployable helix antenna.

b) Explain its advantages and disadvantages (if any).

c) Mention its performance parameters such as physical dimensions, frequency bandwidth, average radiation gain and efficiency.

d) Its practical applications.

2) Since the topic of the paper is dealing with antenna design, it is suggested to add some examples of various antennas. Below are some helpful suggestions.

"A Comprehensive Survey of "Metamaterial Transmission-Line Based Antennas: Design, Challenges, and Applications"", IEEE Access, vol. 8, pp. 144778-144808, 2020.

"On-Chip Antenna Design Using the Concepts of Metamaterial and SIW Principles Applicable to Terahertz Integrated Circuits Operating over 0.6–0.622 THz" International Journal of Antennas and Propagation, Volume 2020, Article ID 6653095, 9 pages, https://doi.org/10.1155/2020/6653095.

"A Comprehensive Survey on "Various Decoupling Mechanisms with Focus on Metamaterial and Metasurface Principles Applicable to SAR and MIMO Antenna Systems"", IEEE Access, vol. 8, pp. 192965-193004, 2020.

3) In the second paragraph of section 3 (theoretical background) authors have mentioned the various geometrical parameters of the helix antenna, but in Fig.1 only three of these parameters have been mentioned. So, to better understand readers please mention all parameters similar to “D, S, and A”.

4) Fig.1 shows the overall geometry of helical antennas, please describe how to optimize the geometrical parameters mentioned in the second paragraph of section 3?

5) Can authors provide more discussion in detail regarding helix antenna radiation modes displayed in Fig.2?

6) Authors have calculated theoretical directivities, so can authors plot these results?

7) In Figs. 3 and 4 authors have shown the model with infinite GP and with CubeSat GP, the advantages and disadvantages of these two models can be discussed and summarized in a table.

8) Fig. 4 describes the ground plane models (square, cylindrical, and conical), can authors please summarize the advantages and disadvantages of the helix antennas with these GP models in a table?

9) In the conclusion section please add some numerical results.

10) Reference part can be improved by a proper extension as per above-mentioned suggestions.

Round 2

Reviewer 2 Report

The reviewers' comments and concerns were carefully considered by properly addressing them during revision. The modified version presents a higher quality work than its initial one, so there are no more technical comments from this reviewer's point of view.